

# Rapid response of arbuscular mycorrhizal fungal communities to short-term fertilization in an alpine grassland on the Qinghai-Tibet Plateau

Xingjia Xiang[1,2], Sean M. Gibbons[3], Jin-Sheng He[4,5], Chao Wang[4], Dan He[1], Qian Li[5], Yingying Ni[1] and Haiyan Chu[1]

[1] State Key Laboratory of Soil and Sustainable Agriculture, Institute of Soil Science, Chinese Academy of Sciences, Nanjing, China
[2] University of Chinese Academy of Sciences, Beijing, China
[3] Department of Biological Engineering, Massachusetts Institute of Technology, Cambridge, Massachusetts, USA
[4] Department of Ecology, College of Urban and Environmental Sciences, Peking University, Beijing, China
[5] Northwest Institute of Plateau Biology, Chinese Academy of Sciences, Xining, Qinghai, China

Corresponding author
Haiyan Chu, hychu@issas.ac.cn

## ABSTRACT

**Background:** The Qinghai-Tibet Plateau (QTP) is home to the vast grassland in China. The QTP grassland ecosystem has been seriously degraded by human land use practices and climate change. Fertilization is used in this region to increase vegetation yields for grazers. The impact of long-term fertilization on plant and microbial communities has been studied extensively. However, the influence of short-term fertilization on arbuscular mycorrhizal fungal (AMF) communities in the QTP is largely unknown, despite their important functional role in grassland ecosystems.

**Methods:** We investigated AMF community responses to three years of N and/or P addition at an experimental field site on the QTP, using the Illumina MiSeq platform (PE 300).

**Results:** Fertilization resulted in a dramatic shift in AMF community composition and NP addition significantly increased AMF species richness and phylogenetic diversity. Aboveground biomass, available phosphorus, and $NO_3^-$ were significantly correlated with changes in AMF community structure. Changes in these factors were driven by fertilization treatments. Thus, fertilization had a large impact on AMF communities, mediated by changes in aboveground productivity and soil chemistry.

**Discussion:** Prior work has shown how plants often lower their reliance on AMF symbioses following fertilization, leading to decrease AMF abundance and diversity. However, our study reports a rise in AMF diversity with fertilization treatment. Because AMF can provide stress tolerance to their hosts, we suggest that extreme weather on the QTP may help drive a positive relationship between fertilizer amendment and AMF diversity.

## INTRODUCTION

The Qinghai-Tibetan Plateau (QTP), also known as the 'Roof of the World' or the 'Third Pole,' is famous for its extreme weather and plays a crucial role in providing ecological stability and climate buffering to the region (*Dong et al., 2010*). About 85% of the QTP is grassland, accounting for more than 30% of all grassland area in China. However, this ecosystem is being degraded by long-term livestock overgrazing and climate change, which have created a series of social, environmental, and economic crises (*Wen et al., 2013*). Chemical fertilizers (i.e. N, P) are widely used in this region to increase vegetation yields for grazing (*Beauchamp, Trevors & Paul, 1989*; *Hacker, Toole & Melville, 2011*; *Zhou et al., 2015*). N is usually considered the major limiting element for plant and microbial growth in most terrestrial ecosystems (*Elser et al., 2007*; *LeBauer & Treseder, 2008*). Recently, it has been argued that phosphorous is equally important in many terrestrial ecosystems (*Vitousek et al., 2010*; *Harpole et al., 2011*). Nitrogen addition has a marked effect on soil phosphorus availability, as N addition induces a shift in soil nutrient limitation from N to P and vice versa (*Lü et al., 2013*).

Fertilization greatly alters soil N and P cycling, which has a profound impact on structure and function of grassland ecosystems. Empirical studies have shown how fertilization significantly reduces plant root biomass (*Bloom, Chapin & Mooney, 1985*) and grass species diversity (*Bai et al., 2010*; *Clark & Tilman, 2008*). Forb biomass generally increases with fertilizer amendment (*Heil & Bruggink, 1987*; *Herron et al., 2001*; *Zhang et al., 2011*). Fertilization is known to alter successional trajectories (*Tilman, 1987*) and promote the establishment of invasive plant species (*Huenneke et al., 1990*). Nitrogen and phosphorus addition have significantly altered soil organic carbon in arctic tundra (*Mack et al., 2004*) and soil carbon fractions in an alpine meadow (*Li et al., 2014*). Long-term fertilization has been associated with significant decreases of soil microbial biomass and bacterial diversity, and is known to drive shifts in bacterial community composition in grassland ecosystems (*Lovell, Jarvis & Bardgett, 1995*; *Ramirez et al., 2010*; *Allison et al., 2013*) and agriculture soils (*Enwall et al., 2007*; *Yuan et al., 2012*; *Sun et al., 2015*). Soil fungal communities are also altered by fertilization in forest (*Liu et al., 2012a*), agricultural (*Avio et al., 2013*), and grassland ecosystems (*Klabi et al., 2015*). Arbuscular mycorrhizal fungal (AMF) communities were significantly altered by long-term balanced fertilization in agriculture (*Lin et al., 2012*) and grassland soils (*Liu et al., 2012b*). Despite a substantial body of work on how long-term fertilization affects soil biogeochemistry and microbial community structure, the response of AMF communities to short-term fertilization (e.g. three years in this study) is not well understood.

AMF form mutualistic symbioses with > 80% of land plants, providing their hosts with nutrients and stress tolerance in exchange for photosynthate (*van der Heijden et al., 1998a*; *Moora et al., 2004*; *Smith & Read, 2008*). AMF communities are structured largely by plant communities (*Johnson et al., 2004*; *Börstler et al., 2006*; *Hausmann & Hawkes, 2009*), as most AMF taxa show some level of host-preferences (*Fitter, 2005*; *Öpik et al., 2009*). The correlation between plant diversity and AMF richness has been reported as positive (*Landis, Gargas & Givnish, 2004*; *Wu et al., 2007*), non-significant (*Börstler et al., 2006*;

*Öpik et al., 2008*) or negative (*Antoninka, Reich & Johnson, 2011*). AMF-plant mutualisms depend on soil nutrient availability and weather conditions. Under high-nutrient conditions, plants obtain enough nutrients without their AMF partners (*Lin et al., 2012*), which can lead to a reduction in carbon allocated to the roots (*Brouwer, 1983*) and an overall drop in AMF abundance (*Johnson, 2010*). Some researchers have argued that the relationship between plants and AMF may even shift from mutualism to parasitism in a nutrient-rich environment (*Johnson, 1993*; *Chu et al., 2016*). In addition to nutrient acquisition, AMF can help buffer their plant hosts from environmental stress (e.g. cold and drought) by causing shifts in host physiology (*Miransari et al., 2008*). AMF colonization has been shown to increase under stressful conditions, which is thought to improve plant performance by inducing increased production of secondary metabolites, higher enzymatic activities, and greater plasma membrane permeability. (*Bunn, Lekberg & Zabinski, 2009*; *Chen et al., 2013*). We designed the following study to investigate the response of AMF communities to short-term fertilization in an alpine grassland on the QTP. In particular, we addressed two main questions: (i) how does short-term fertilization affect AMF community structure on QTP; and (ii) what environmental parameters shape AMF community structure following fertilization?

## MATERIALS AND METHODS

### Site selection and soil sampling

The fertilization addition experiment started in 2011 (37°37′N, 101°12′E; 3,220 m), at the Haibei Alpine Grassland Ecosystem Experimental Station (QTP station), which was established in 1976 (northeast of QTP; 37°29′–37°45′N, 101°12′–101°23′E). The Haibei Alpine Grassland Ecosystem Experimental Station provided us with the necessary permits to carry out this work. The field sites were strictly vetted based on our selection standards: Mat-Cryic Cambisols soil type, vegetation type, and environmental conditions. The area has a typical plateau/continental climate, with a long winter. The average annual temperature is −1.7 °C, with a maximum temperature of 27.6 °C and a minimum temperature of −37.1 °C.

Soils were collected on the 12[th] of August, 2014, at QTP Station. The experimental plots were constructed in randomized block design and the plot was a 6 × 6 m square. Our study included four treatments, with five replicates each: Control (without fertilization); N (100 kg N ha$^{-1}$ yr$^{-1}$); P (50 kg P ha$^{-1}$ yr$^{-1}$); and NP (100 kg N ha$^{-1}$ yr$^{-1}$ and 50 kg P ha$^{-1}$ yr$^{-1}$) (Table S1). Our fertilizer amendment amounts fall within an intermediate range of what other studies report (*Sun et al., 2015*; *Zeng et al., 2015*). The fertilizers were urea and $Ca(H_2PO_4)_2$, applied on the first days of June, July and August (i.e. during the main growing season). In each plot, soils were collected from four corners of the plot (1 m from edge of plot to avoid edge effects and at a depth of 0–10 cm) and then mixed together into one sample. The soils were kept in a cooler and shipped refrigerated to the lab as quickly as possible. The samples were completely mixed within each bag and sieved to remove stones, and then divided into three parts: one part was for biogeochemical analysis and was stored at 4 °C; the second part was stored at −20 °C for DNA extraction and the third part was placed in long-term storage at −40 °C.

## Vegetation and soil properties analyses

Aboveground Net Primary Productivity (ANPP) were assessed in four $0.5 \times 0.5$ m areas, at the corners of soil plots. Aboveground portion of biomass was collected from a plot using clipper. Plants were grouped visually into four categories: Gramineae, Sedge, Legume and Forb. In each plot, the roots from three replicate soil cores which were collected using a 7 cm diameter auger were used to estimate belowground root biomass. The dry biomass for aboveground sections and belowground roots (after washing) were calculated after 48 h in the 65 °C oven. Soil pH was measured after shaking a soil water suspension (1:5 wt/vol) for 30 min and soil moisture (SM) was measured gravimetrically. The classical methods were applied for measuring soil available phosphorus (AP, *Ståhlberg, 1980*), total phosphorus (TP, *Bowman, 1988*), total carbon (TC) and total nitrogen (TN) (*Walkley & Black, 1934*). Soil dissolved organic C (DOC) and total dissolved N (TDN) and mineral nitrogen were extracted by adding 50 ml of 0.5 M $K_2SO_4$ to 10 g fresh soil, shaking for 1 h and then vacuum filtering through glass fiber filters (Fisher G4, 1.2 μm pore space). Ammonium ($NH_4^+$) and nitrate ($NO_3^-$) contents in the extracts were determined colorimetrically by automated segmented flow analysis (Bran + Luebbe AAIII, Germany) using the salicylate/dichloroiso- cyanuric acid and cadmium column/sulphanilamide reduction methods, respectively. DOC and TDN were determined using a TOC-TN analyzer (Shimadzu, Kyoto, Japan). Dissolved organic N (DON) was calculated as follows: $DON = TDN - NH_4^+ - NO_3^-$. Biogeochemical data are shown in Table S2.

## Soil DNA extraction

DNA extractions were carried out on 0.5 g soil according to the manufacturer's instructions (FastDNA® SPIN Kit for soil, MP Biomedicals, Santa Ana, CA). The extracted DNA was dissolved in 50 μl TE buffer, quantified by NanoDrop ND-1000 (Thermo Scientific, USA) and stored at −20 °C.

## PCR and amplicon library preparation

Primers AMV4.5NF and AMDGR (*Lumini et al., 2010*) were used to amplify soil 18S rRNA gene fragments for the Illumina MiSeq platform (PE 300) at Novogene (Beijing, China). PCR was carried out in 50-μl reaction mixtures containing each deoxynucleoside triphosphate at a concentration of 1.25 mM, 1 μl of forward and reverse primers (20 μM), 2 U of Taq DNA polymerase (TaKaRa, Japan), and 50 ng of DNA. The following cycling parameters were used: 35 cycles (95 °C for 45 s, 56 °C for 45 s, and 72 °C for 45 s) were performed with a final extension at 72 °C for 10 min. Triplicate reaction mixtures per sample were pooled together and purified using an agarose gel DNA purification kit (TaKaRa), and quantified using NanoDrop ND-1000 (Thermo Scientific, Waltam, MA, USA) ranging from 44.2 to 82.7 ng/μl. The bar-coded PCR products were pooled in equimolar amounts (10 pg for each sample) before sequencing.

## Processing of sequence data

Sequences were merged by FLASH (*Magoc & Salzberg, 2011*) and then processed using Quantitative Insights Into Microbial Ecology (QIIME; http://www.qiime.org/)

(*Caporaso et al., 2010*). Poor-quality sequences (below an average quality score of 25) and short sequences (< 200 bp) were removed. Sequences were clustered into Operational Taxonomic Units (OTUs) using a 97% identity threshold (default QIIME settings) by UCLUST (*Edgar, 2010*) and all singleton OTUs were deleted. Chimera filtering was also performed to remove sequencing errors with USEARCH tool (version 1.8.0) in QIIME. The most abundant sequence within each cluster was selected as the representative sequence for that OTU. The representative sequences were checked against the MaarjAM AMF database (*Öpik et al., 2010*; http://maarjam.botany.ut.ee/). We rarified the abundance matrix to 1,000 sequences per sample to obtain normalized relative abundances.

## Statistical analysis

Phylogenetic diversities (PD) were estimated by Faith's index (*Faith, 1992*), providing an integrated index of the phylogenetic breadth across taxonomic levels. Pearson correlations were calculated between AMF alpha-diversity and biogeochemical properties using SPSS 20 for Windows. A neighbor-joining tree (MEGA 6; *Tamura et al., 2013*) was built for phylogenetic assignment by aligning dominant AMF OTU sequences (relative abundance > 1%) with known reference sequences. The response ratio (RR) was used to quantify significant responses of OTUs to fertilization. Only those OTUs detected in more than 3 replicates of each treatment were analyzed. The 95% confidence interval (CI) = $rr_i \pm 1.96 \times \sqrt{V_i}$, where $rr_i = \ln(\bar{x}_i/\bar{y}_i)$ ($i = 1 \ldots$ n), $\bar{x}$ is the mean OTU read number in fertilization samples and $\bar{y}$ is the mean OTU read number in control samples; the variance ($V_i$) is, $V_i = \frac{s_{x_i}^2}{m_{x_i}\bar{x}_i^2} + \frac{s_{y_i}^2}{m_{y_i}\bar{y}_i^2}$ ($i = 1 \ldots$ n), where $s$ is the SD of OTU $i$ in control and fertilization samples and $m$ is the abundance of OTU $i$ in control and fertilization samples. The *rr-significance* value was calculated using the equation: $rr - significance = (rr_i + 1.96\sqrt{V_i}) \times (rr_i - 1.96\sqrt{V_i})$. If the 95% CI of a response variable overlaps with zero (*rr-significance* < 0), the RR at fertilization is not significantly different from the control. If *rr-significance* > 0, the RR is significant (*Luo, Hui & Zhang, 2006*; *Xiang et al., 2015*). Non-metric multidimensional scaling (NMDS) based on Bray-Curtis dissimilarity (calculated from the relative abundance matrix), and Analyses of Similarities (ANOSIM) (*Clarke, 1993*) were performed to compare community composition in different treatments in the *vegan* package 2.0–10 (*Dixon, 2003*) of R v.3.1.0 (R Development Core Team, Vienna, Austria). Mantel tests were used to identify biogeochemical factors that significantly correlated with community composition. Multivariate regression trees (MRT) were constructed to display important relationships between AMF community and biogeochemical variables in the *mvpart* package 1.2–6 (*De'ath, 2002*).

# RESULTS

## General sequencing information

We obtained a total of 380,203 quality-filtered fungal sequences across all soil samples for the primer pair AMV4.5NF/AMDGR: 12.14% were classified as Glomeromycota (AMF). A total of 46,140 quality-filtered AMF sequences were identified across all the

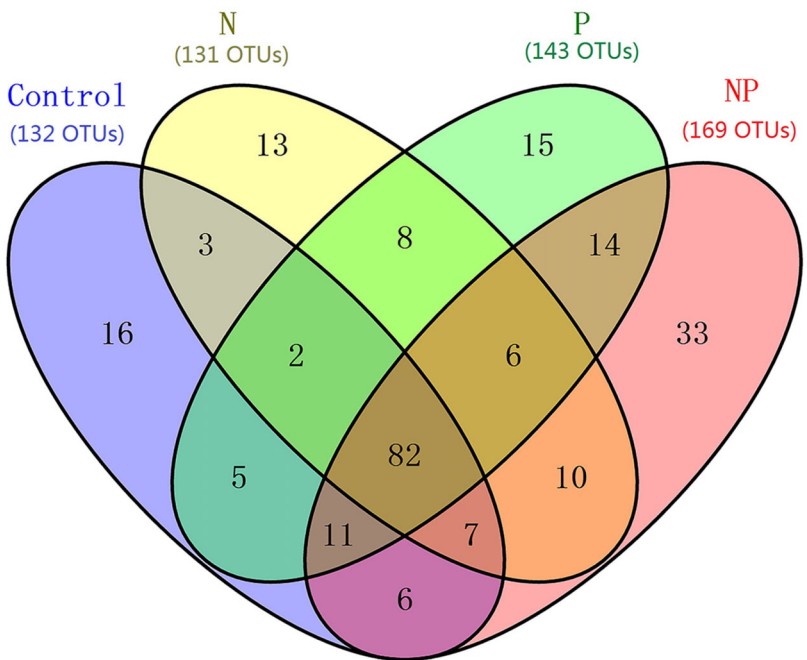

**Figure 1 Venn diagram showing the co-occurence of the OTUs among samples from different treatments at the QTP station.** Numbers in parentheses indicate total OTUs in each treatment group, and numbers inside the Venn diagram indicate unique and shared OTUs. OTU, operational taxonomic unit.

entire data set, ranging from 1,088–4,313 sequence reads per sample (mean = 2,307). The neighbor-joining tree showed that the dominant OTUs (relative abundance > 1%) grouped into *Claroideoglomeraceae* (30.1%), *Gigasporaceae* (23.9%), *Glomeraceae* (23.4%), *Diversisporaceae* (12.6%) and *Acaulosporaceae* (9.1%) families (Figs. S1 and S2). In addition, *Ambisporaceae, Archaeosporaceae, Pacisporaceae* were present in most soils but at relatively low abundances, indicating a good coverage of the Glomeromycota.

## Arbuscular mycorrhizal fungal alpha-diversity

After rarefaction to an equal sequencing depth per sample (1,000), we found a total of 231 distinct AMF OTUs across all samples (97% similarity), 35.5% of which (82) were found in all treatments. There were 132, 131, 143 and 169 AMF OTUs in Control, N, P and NP treatments, respectively. In addition, there were unique OTUs within each treatment, especially for NP plots, where the unique OTUs accounted for 19.5% of the 169 distinct OTUs observed (Fig. 1). Soil AMF alpha-diversity (i.e. OTU richness and phylogenetic diversity) was calculated at a depth of 1,000 randomly selected sequences per sample. Compared to control soils, NP addition significantly increased AMF OTU richness and phylogenetic diversity in this study (Fig. 2). Of all the biogeochemical characteristics examined, both OTU richness and phylogenetic diversity were positively correlated with $NO_3^-$, DON, AP, graminoid biomass, forb biomass and TB and negatively correlated with legume biomass ($P < 0.05$ in all cases; Table S3).

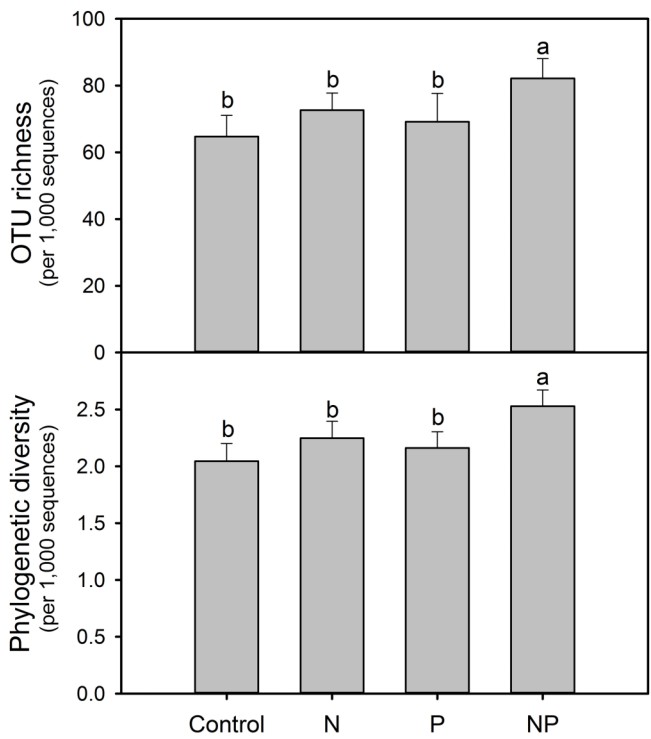

**Figure 2 Soil AMF alpha-diversity (OTU richness and phylogenetic diversity) calculated at a rarefaction depth of 1,000 randomly selected sequences per sample in soils across different treatments.** Different letters represent significant differences from Tukey's HSD comparisons ($P < 0.05$). Error bars denote standard deviation. OTU, operational taxonomic unit.

## Arbuscular mycorrhizal fungal community composition

Compared to control plots, NP addition significantly decreased the relative abundance of *Gigasporaceae* and increased the relative abundance of *Glomeraceae*. The relative abundance of *Diversisporaceae* showed significant increase and the relative abundance of *Acaulosporaceae* showed significant decrease in N, P and NP plots relative to control soils (Fig. 3). RR were calculated in order to identify significant differences across control and fertilization samples (Fig. 4). In the family *Diversisporaceae*, all OTUs showed significant increases in abundance, and for *Acaulosporaceae*, all the OTUs showed significant decreases in abundance in fertilization plots. In the family *Gigasporaceae*, all OTUs in P and NP plots showed negative responses relative to the control, and for the family *Glomeraceae*, all OTUs showed positive responses in NP plots relative to the control.

AMF community composition was significantly influenced by short-term fertilization (Fig. 5; Table S4). Mantel tests showed that the AMF community composition was significantly correlated with AP, $NO_3^-$, TP, graminoid biomass, legume biomass, forb biomass and total aboveground biomass (TB) ($P < 0.05$ in all cases; Table S5). In addition, MRT analysis was used to investigate the effects of biogeochemical variables on AMF communities. The MRT model explained 37.0% of detected variation in AMF community composition. The important factors were AP,

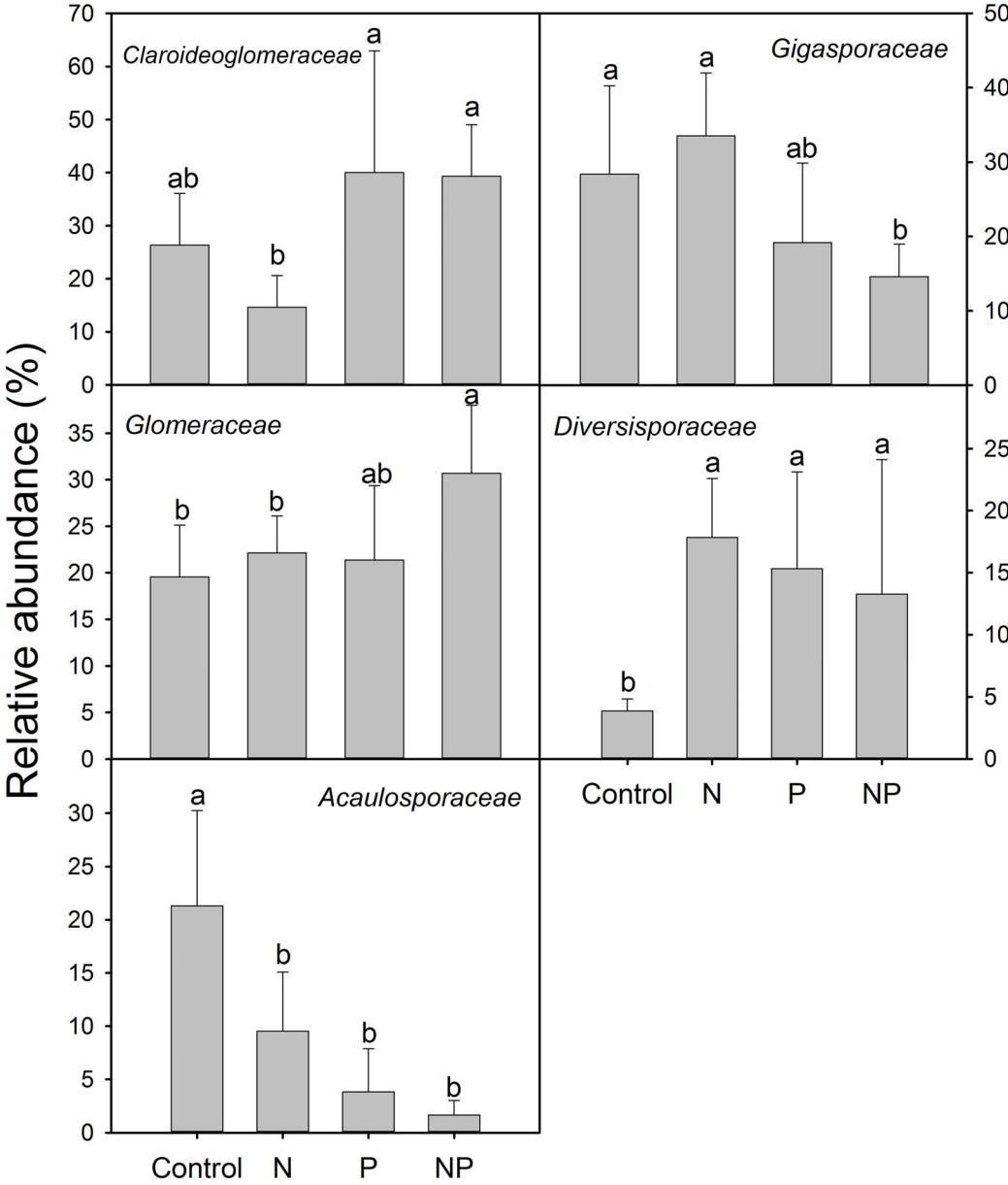

**Figure 3 Relative abundances of dominant AMF families across control and fertilization (N, P and NP) treatments at the QTP station.** Error bars denote standard deviation calculated from five samples; different letters represent significant differences from Tukey's HSD comparisons ($P < 0.05$).

TB and legume biomass, which accounted for 20.5, 8.4 and 7.1% variation of AMF community composition, respectively (Table S6). Both AP and TB showed significant Pearson correlations with the relative abundance of *Claroideoglomeraceae*, *Gigasporaceae* and *Acaulosporaceae*; Both soil DON and forb biomass showed significant Pearson correlations with the relative abundance of *Glomeraceae* and *Acaulosporaceae*; TP showed significant Pearson correlations with the relative abundance of *Gigasporaceae* and legume biomass showed significant Pearson correlations with the relative abundance of *Glomeraceae* ($P < 0.05$ in all cases; Table S7).

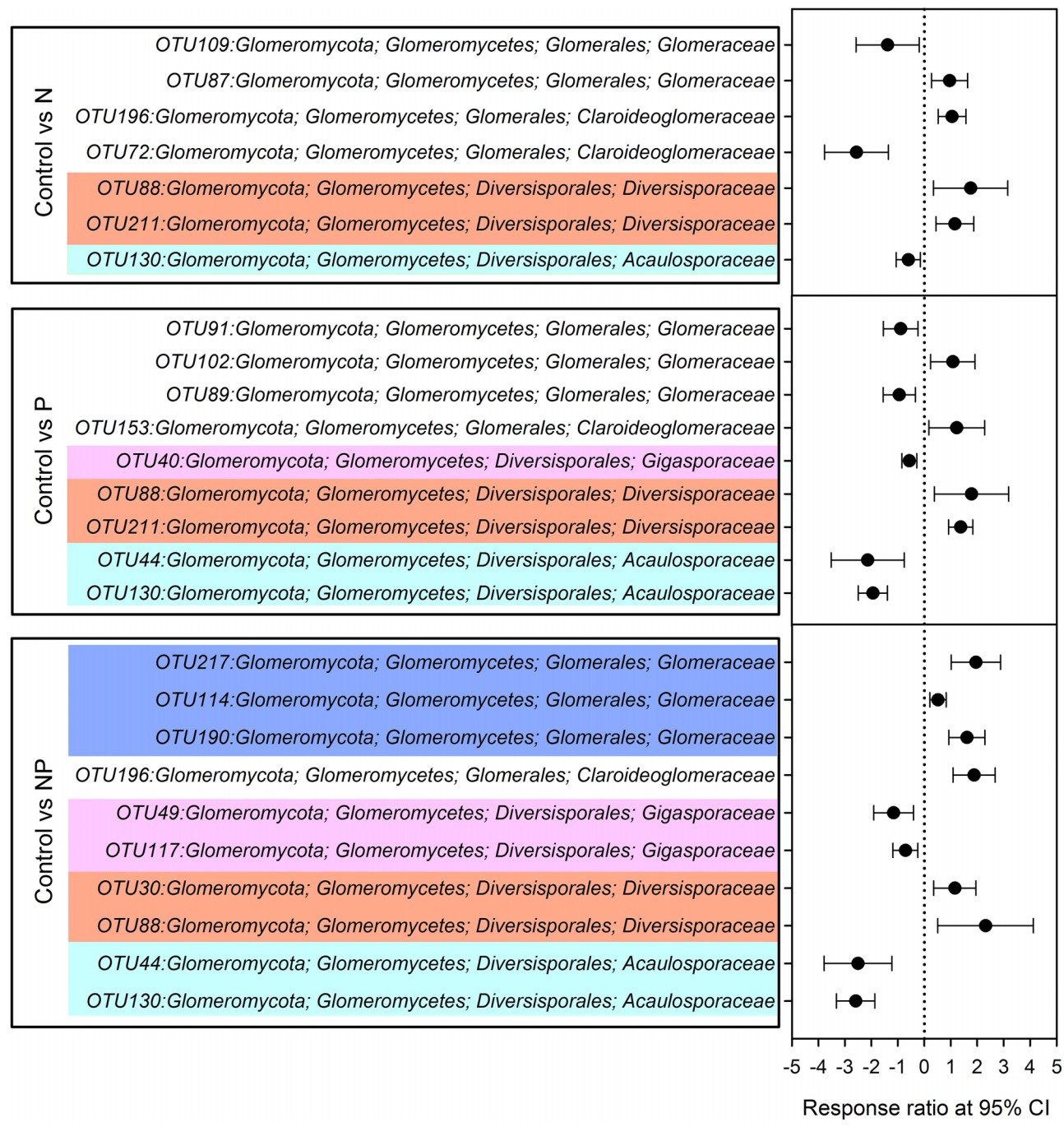

**Figure 4 Significant responses of AMF Operational Taxonomic Units (OTUs) after N, P and NP fertilization relative to control plots.** Significance was determined using a 95% confidence interval (CI), according to RR method.

## DISCUSSION

Chemical fertilizers (i.e. N, P) are common remediation tools that help maximize biomass production and prevent desertification in grassland ecosystems (*Beauchamp, Trevors & Paul, 1989*; *Hacker, Toole & Melville, 2011*). Despite a substantial body of work on how long-term fertilization affects vegetation and belowground microbial communities, effects

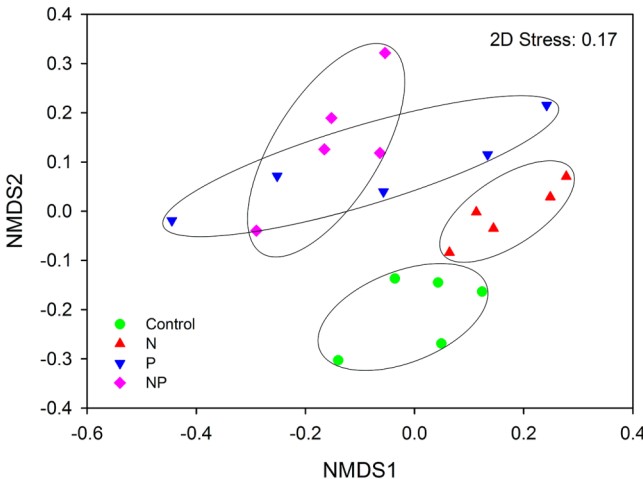

**Figure 5** NMDS plot of AMF community composition across control and fertilization (N, P and NP) treatments at the QTP station.

of short-term fertilization on soil microbes in alpine grasslands remains largely unexamined. Our study was performed at a well-managed field station, where site conditions were well documented prior to our fertilization study. We examine the effects of short-term fertilization on soil AMF communities, as AMF are ubiquitous and play a major role in grassland ecosystem function (*van der Heijden et al., 1998b*; *Johnson et al., 2004*; *Gianinazzi et al., 2010*). The most dominant AMF family in this ecosystem was *Claroideoglomeraceae. Glomeraceae* is usually predominant in agriculture (*Oehl et al., 2005*; *Hijri et al., 2006*; *Lin et al., 2012*), grassland (*Maherali & Klironomos, 2012*; *Zangaro et al., 2013*) and forest ecosystems (*Helgason et al., 2007*; *Xiang et al., 2015*; *Chu et al., 2016*) around the world, which highlights the unique nature of the QTP. Short-term fertilization significantly altered soil properties (i.e. AP, TP, $NO_3^-$ and DON) and aboveground biomass, suggesting rapid response of biogeochemical properties to fertilization. We found that short-term fertilization resulted in a dramatic shift of AMF communities, which was corroborated by results from other studies in grassland (*Porras-Alfaro et al., 2007*; *Liu et al., 2012b*; *Chen et al., 2014*) and agriculture soils (*Antoninka, Reich & Johnson, 2011*; *Lin et al., 2012*; *Avio et al., 2013*), which confirms how sensitive AMF communities are to soil nutrient levels. AMF communities showed strong correlations with AP, $NO_3^-$ and aboveground biomass, suggesting that the effects of fertilization on AMF communities are mediated by changes in vegetation structure and soil chemistry.

Generally, fertilized plants can obtain enough nutrients from soil without their AMF partners, which lowers the reliance of the plant community on AMF symbioses (*Johnson, 1993*). Higher soil nutrient levels have been shown to decrease root biomass (*Bloom, Chapin & Mooney, 1985*) and increase aboveground biomass (i.e. light; *Hautier, Niklaus & Hector, 2009*). Thus, fertilized plants reduce their allocation of belowground carbon (*Brouwer, 1983*) and thereby reduce the abundance and diversity of AMF mutualists (*Johnson, 2010*). Conversely, in our study NP fertilization led to a slight increase in root biomass and a marked increased AMF alpha-diversity. Despite reduced reliance on AMF mutualists for nutrients in the presence of fertilizers, plants may

still rely on AMF-mediated stress tolerance to deal with the extreme weather conditions (e.g. cold) common to the QTP (*Chen et al., 2013*). Fertilizer-induced changes in the AMF communities could reflect selection for mutualisms that promote stress tolerance. For example, previous studies have shown that certain AMF, primarily within the family of *Glomeraceae*, provide their hosts with improved cold tolerance (*Zhu, Song & Xu, 2010*; *Chen et al., 2013*). We found that the relative abundance of *Glomeraceae* significantly increased in response to NP treatment.

NP fertilization led to a two-fold increase in grass biomass, with a slight increase in root biomass. Thus, the absolute amount of photosynthate available to AMF may have remained constant, or even increased. We propose the following model for fertilization in the QTP: NP fertilization increases grass biomass; higher overall photosynthesis allows for a sufficient carbon flux to the root system to support AMF mutualists; AMF provide their host with improved stress tolerance. As belowground carbon allocation increases, competition between AMF is reduced and rare species are better able to persist.

In addition, several studies have demonstrated that fertilization increases the occurrence of non-native plant species (*Huenneke et al., 1990*; *Hobbs et al., 1988*; *Ostertag & Verville, 2002*). AMF taxa show a certain degree of host preference (*Vandenkoornhuyse et al., 2003*; *Fitter, 2005*; *Öpik et al., 2009*; *Liu et al., 2012b*), which might in turn increase AMF diversity in the presence of non-native plants (*Lekberg et al., 2013*).

## CONCLUSIONS

In conclusion, our study shows that fertilization increases AMF community diversity in the QTP, which is the opposite of what has been found in several other studies. We suggest that both soil nutrients and stress tolerance should be considered in order to thoroughly evaluate the effect of fertilization on AMF communities on QTP. This work helps to build a more complete picture of how AMF communities respond to fertilization across different ecosystems. Future work should focus on more detailed plant data and longitudinal sampling (short, intermediate and long-term) following fertilization to identify the feedbacks between plant and AMF communities over time.

## ACKNOWLEDGEMENTS

We thank Ms. Kaoping Zhang, Mr. Yuntao Li and Dr. Ruibo Sun from Institute of Soil Science, Chinese Academy of Sciences, for assistance in soil sampling. We also thank Dr. Yu Shi from Institute of Soil Science, Chinese Academy of Sciences, for useful discussion.

### Funding

This work was supported by the National Program on Key Basic Research Project (973 Program, Grant #2014CB954002), the Strategic Priority Research Program (Grant #XDB15010101) of the Chinese Academy of Sciences, and the National Natural Science Foundation of China (41071121). The funders had no role in study design, data collection and analysis, decision to publish, or preparation of the manuscript.

## Grant Disclosures

The following grant information was disclosed by the authors:

National Program on Key Basic Research Project (973 Program): #2014CB954002.

Strategic Priority Research Program of the Chinese Academy of Sciences: #XDB15010101.

## Competing Interests

The authors declare that they have no competing interests.

## Author Contributions

- Xingjia Xiang performed the experiments, analyzed the data, contributed reagents/materials/analysis tools, wrote the paper, prepared figures and/or tables, reviewed drafts of the paper.
- Sean M. Gibbons analyzed the data, contributed reagents/materials/analysis tools, wrote the paper, reviewed drafts of the paper.
- Jin-Sheng He conceived and designed the experiments, reviewed drafts of the paper.
- Chao Wang contributed reagents/materials/analysis tools, reviewed drafts of the paper.
- Dan He performed the experiments, contributed reagents/materials/analysis tools, reviewed drafts of the paper.
- Qian Li contributed reagents/materials/analysis tools, reviewed drafts of the paper.
- Yingying Ni contributed reagents/materials/analysis tools, reviewed drafts of the paper.
- Haiyan Chu conceived and designed the experiments, analyzed the data, reviewed drafts of the paper.

## Data Deposition

GenBank: SRP072405.

## Supplemental Information

Supplemental information for this article can be found online at http://dx.doi.org/10.7717/peerj.2226#supplemental-information.

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
