# Peer review of "Rapid response of arbuscular mycorrhizal fungal communities to short-term fertilization in an alpine grassland on the Qinghai-Tibet Plateau"

_PeerJ, doi:10.7717/peerj.2226_

## Round 0.1 · original submission · Major Revisions

Please revise your manuscript to address the questions and comments by the reviewers. I think more detail in the methods and results would help clarify your experimental methods for Reviewer 1. Consider moving the supporting methods to the main methods section because they are not extensive. The Results section is also very brief and it would benefit readers to provide a summary of the treatment effects presented in Table S1 in the main manuscript. Because your findings are different from some other studies of fertilized plants, I think it is important to address Reviewer 2's question about how the concentration of fertilizers applied and the resulting concentrations of N and P in the soil compare to these other studies. Please also fix the font size of "Gramineae" in Table S5.

Reviewer 1 ·

Basic reporting

The paper by Xiang et al describes the response of AMF to short therm fertilization in an peculiar system such as grasslands in the Qinghai-tibet Plateau. It is generally well written.
Introduction could be improved:
-line 88: explain better how weather conditions should affect AMF communities
In general: since there is few knowledge on effect of shirt term fertilization, please stress more the novelty of the work. even in the discussion.

Experimental design

This is the section where I saw more critical points that should be addresses for the acceptance for publication:
-line 98: which were the initial condition of soils? were they all in the same condition before the startingof the fertilization treatment? More details are needed here and it would be useful also in the first part of the discussion section.
From table S1 it seems that there are no differences in the N content among the four level of treatment. please explan.
-line 104: authors refers to “split block design”. But this experimental design requires two treatments while in the paper there is only one (i.e. fertilization). Maybe authors want to refer to “randomized block design”? More clarification should be given: this is a critical point
-line 105: treatment is only one (= fertilization) with 4 levels. Please correct here and through all the MS and also in the figure captions
-line 108: which is the real dimension of the plot? 4x4m or 6x6m?
-line 112: is -20° or -80°?
-line 118: why only three replicates instead of four (as for the above ground biomass?)
-line 116-120: the description of the vegetation sampling is lacking of important details. In the text and in tables authors describe graminoid biomass, legume biomass,… but there is no mention at all in the methodology. Please improve and describes how you measure them.
-line 134: which was the range of the quantified DNA in the samples?
- line 135: which were the PCR condition to melt bar coded with pcr products? Specifiy the equimolar amount of DNA needed
-line 148: why did you chose to rarefy to 1,000 sequences? How did you chose this number? (i.e. median numbers of reads?) I think it would be appropriate to repeat analyses even without normalization, to check that normalization does not affect the final results.
-line 152: please specify which analyses you performed.
-line 153: what is response ratio? How is calculated? What is its meaning? Please specify
-line 155: NMDS was calculated on what? Matrixes of relative abundances? Presence/absence?

Validity of the findings

Please improve the discussion section discussing the results of the biogeochemical values and how theay affect AMF community.
- line 165: authors should detail how many sequences they found in total and which was the % of AMF sequences compared to the total.
- line 175: the decrease is significant only compared to Control. please specify better
line 177-178: rephrase better. significant differences were among which levels of the treatment?
-line 186: you did not mention them in the M&M section
-line 189: please provide a picture of the MRT model or provide a table from wich to check the values,
-line 195:which families?
-line 197: alpha diversity should be described (and discussed) BEFORE the community structure in the text. please follow this suggestion
-line 198: here authors refer to VT. In the previous paragraph authors refer to OTU. Please be consistent. Even in fig 3 I think it would be more useful to read the number of VT instead of OTU: it makes it more comparable to the literature. Pleas improve
line 198 - authors comment the phylogentic diversity: but how do they measure it? please improve M&M

-please provide a phylogenetic tree where to check the phylogeny of your OTUS

line 209: "short term fertilization"
line 235-237: please rephrase
line 247: authors refer to host preference but in this research host plant is not taken in account

Additional comments

line 55: start a new paragraph
line 93: "fertilization affect AMF community structure in QTP"
line 171: OTU and not genera. Delete dominant.
line 173: in fig S1 same colors are used in the legend (i.e violet). please revise
line 184: start a new paragraph
Fig 1: which are the numbers in parentheses?
Fig 2: specify form how many samples the standard deviation comes from
Fig 3: specify the treatment and the levels of treatments. captions should be self explaining. Do it also for fig 4

Pleae revise the supplementary tables: there are some typos (i.e. gramimeae)
Table S1 root biomass is really kg/m2? please check

Reviewer 2 ·

Basic reporting

This article is well written and easy to follow.

Not only is the background sufficient, it does not go beyond the scope of what needs to be presented here. Very nice and streamlined.

Experimental design

Methodological description is very clear.

Validity of the findings

Experimental design is strong and controlled.

Additional comments

General: In this study, the authors examine the effect of fertilization in the Qinghai-Tibet Plateau on AMF community structure and diversity. The authors found, in contrast to previous studies, that AMF diversity increased with fertilization.

Overall, this manuscript was a pleasure to read. The flow was very streamlined, and the authors did not discuss material that was outside of the scope of this study. The manuscript is generally well-written, and the methodological details are clear. In addition, the authors provide a suggested model for their findings at the end of the discussion, without extending much beyond the available data (in this study and those cited). The figures are extremely clear and relevant.

I highly recommend this paper for publication with only minor edits. Please see below.

Grammar: The readability of this article is high. Very easy to follow.

Line 46: Kind of small and nitpicky, but try to reword so that your sentence does not start with a number. You could even write it out as “Eighty-five percent..”

Line 54-55: In a couple of words, how does N addition affect P availability? Because the dominant microbes involved in cycling change?

Line 61-62: I’m confused by what exactly this means: “Fertilization is known to drive successional processes…”. Sounds like they occur only because of fertilization. Do you mean that fertilization speeds up succession? Or alters successional trajectories? Please reword.

Line 105: Can you please state how your fertilizer concentrations compare to previous studies? Is this amount of fertilizer low, moderate, or high?

Line 130: “1 μl of forward and reverse primers”…At what concentration?

Line 242-243: “NP fertilization increases grass biomass..” Just out of interest, do you think that the increase in grass biomass was enough to offset some of the fertilizer effect on the soil system? For instance, competition by two plants for 20 mg/kg fertilizer should be roughly equal to competition by four plants for 40 mg/kg fertilizer in the same space, right?

Not all references are in alphabetical order. See Hijri and Herron.

---

## Round 0.2 · accepted · Accept

I noticed that you uploaded the supporting document as only for review and "not for publication". The Production team will change this to supporting materials for publication.

Reviewer 1 ·

Basic reporting

I think the authors did a very nice job in addressing all issues raised in response to the original submission.

One could still discuss a lot on how rarify the abundance matrix to a given number of sequences per sample to obtain normalized relative abundances: the authors of the papers cited chose the number of sequences on the basis that it was the median.

I still have some concerning on the condition of soils before the experiment started:were they in the same condition. please specify better in the text.

Experimental design

no comments

Validity of the findings

no comments

Additional comments

no comments